# Mortality in Antinuclear Antibody-Positive Patients with and Without Rheumatologic Immune-Related Disorders: A Large-Scale Population-Based Study

**DOI:** 10.3390/medicina61010060

**Published:** 2025-01-02

**Authors:** Uria Shani, Paula David, Ilana Balassiano Strosberg, Ohad Regev, Mohamad Yihia, Niv Ben-Shabat, Dennis McGonagle, Orly Weinstein, Howard Amital, Abdulla Watad

**Affiliations:** 1Department of Medicine B, Zabludowicz Center for Autoimmune Diseases, Sheba Medical Center, Tel-Hashomer 5262100, Israel; uria.shani@gmail.com (U.S.); paula.rdavid@gmail.com (P.D.); ilanabalassi98@gmail.com (I.B.S.); nivben7@gmail.com (N.B.-S.); howard.amital@sheba.health.gov.il (H.A.); 2Sackler Faculty of Medicine, Tel-Aviv University, Tel-Aviv 6997801, Israel; regev.ohad@gmail.com (O.R.); mohamedyi656@gmail.com (M.Y.); 3Section of Musculoskeletal Disease, NIHR Leeds Musculoskeletal Biomedical Research Unit, Leeds Institute of Molecular Medicine, University of Leeds, Chapel Allerton Hospital, Leeds LS7 4SA, UK; d.g.mcgonagle@leeds.ac.uk; 4Department of Geriatrics, Sheba Medical Center, Tel-Hashomer 5262100, Israel; 5Clalit Health Services, Tel-Aviv 4933355, Israel; orlyweinstein@clalit.co.il; 6Faculty of Health Sciences, Ben-Gurion University of the Negev, Ben-Gurion Ave., Beer Sheva 8410501, Israel

**Keywords:** anti-nuclear antibody, mortality, autoimmunity, autoinflammatory

## Abstract

*Background & Objectives*: To explore the potential association between positive ANA serology and all-cause mortality in a large cohort of patients, including those with and without rheumatological conditions and other immune-related diseases. *Material and Methods*: A retrospective cohort study analyzed all-cause mortality among 205,862 patients from Clalit Health Services (CHS), Israel’s largest health maintenance organization (HMO). We compared patients aged 18 and older with positive ANA serology (n = 102,931) to an equal number of ANA-negative controls (n = 102,931). Multivariable Cox regression models were used to assess hazard ratios (HR) for mortality, adjusting for demographic and clinical factors. *Results*: ANA positivity was strongly associated with increased mortality (adjusted HR [aHR] 4.62; 95% CI 4.5–4.7, *p* < 0.001). Significant predictors of mortality included male gender (39.2% vs. 24.4%, *p* < 0.001), older age at testing (72.4 ± 13.0 vs. 50.1 ± 17.3 years, *p* < 0.001), and Jewish ethnicity (89.6% vs. 83.2%, *p* < 0.001). Certain ANA patterns, such as mitochondrial (and dense fine speckled (DFS-AC2)), were highly predictive of mortality, with aHRs of 36.14 (95% CI 29.78–43.85) and 29.77 (95% CI 26.58–33.34), respectively. ANA-positive patients with comorbid rheumatological immune-related disorders (RIRDs) demonstrated a higher survival rate compared to those without such a condition (aHR 0.9, 95% CI 0.86–0.95, *p* < 0.001). This finding remained significant after adjusting for several parameters, including age. *Conclusions*: ANA positivity is associated with increased all-cause mortality, particularly in individuals without rheumatologic disorders, after adjusting for confounders such as age. This may indicate occult malignancies, cardiovascular pathology, or chronic inflammatory states, necessitating more vigilant surveillance

## 1. Introduction

Antinuclear antibodies (ANAs) are autoantibodies that target the nucleus of normal human cells, indicating a breakdown of immunologic tolerance [1]. The detection of ANA is important for diagnosing ANA-associated rheumatic diseases (AARD), including systemic erythematous lupus (SLE) and systemic sclerosis (SS) [2], but also and other autoimmune conditions [3], including autoimmune thyroid disorders [2], highlighting the complex interplay between various components of the immune system.

ANA positivity can appear years before the development of symptomatic autoimmune disease [4]. The prevalence of ANA positivity is high and nonspecific, found in approximately 12–16% of the general U.S. population, with higher occurrences among women, older individuals (ages ≥ 70 years), African Americans, and those of normal weight [4,5]. In China, ANA positivity rates are also high and correlate with sex and age groups [6].

The reasons for elevated ANA levels in healthy and aging individuals are not well understood, but genetic or environmental factors influencing immune regulation and activity may play a role. These antibodies can also be associated with non-autoimmune factors such as chronic diseases including heart disease [7], malignancies [8,9], infections [10], and certain medications [11], although evidence is limited. Notably, inflammation-related exposures such as obesity and smoking have not been linked to higher ANA prevalence in the U.S. population [4].

However, some research has explored potential links between ANA positivity and mortality due to cardiovascular disease (CVD) and cancer [4]. ANA has shown a weak overall link with all-cause mortality, but studies suggest stronger associations with increased mortality from cardiovascular disease and cancer in certain demographic groups. While ANA may predict cardiovascular events, its connection to cancer risk and prognosis remains unclear, with mixed evidence [1,5,9,12,13]. Unlike cardiovascular diseases and cancer, the impact of chronic inflammatory conditions on all-cause mortality in ANA-positive patients has been less thoroughly investigated.

Due to the limited and often contradictory existing literature, this study aimed to explore the potential association between positive ANA serology and all-cause mortality in a large cohort of patients, including those with and without rheumatological conditions and other immune-related diseases.

## 2. Material and Methods

Study design and population

A retrospective study was conducted in Israel from 1 January 2000 to 31 December 2022, with data recording commencing in 2002. Inclusion criteria included all individuals aged 18 or older with positive ANA serology (i.e., the test group), who were then compared to ANA-negative patients (i.e., the control group). The Ethical Committee of Clalit Health Service (CHS) granted approval for the study, which was exempted from obtaining informed consent.

Data collection

We utilized data from the electronic database of Clalit Health Services (CHS), the largest health maintenance organization (HMO) in Israel [14]. The CHS serves over 4.5 million patients across the country. Under the Israeli National Health Insurance Act of 1994, CHS provides comprehensive healthcare coverage to all citizens. Enrolment in one of the four HMOs is mandatory for every Israeli, ensuring access to healthcare services and insurance coverage regardless of gender, age, health status, or other factors. The extensive CHS database consolidates patient medical records from various healthcare providers, including diagnostic visits, pharmaceutical interventions, in-office procedures, laboratory test results, imaging studies, and comprehensive summaries of hospital encounters such as outpatient clinic visits, emergency department visits, and inpatient discharge records.

Outcomes and definitions

The primary outcome of this study was to compare all-cause mortality rates between ANA-positive and ANA-negative patients. ANA positivity was defined as a titer of ≥1:40, measured by an enzyme-linked immunosorbent assay (ELISA) on at least two occasions. In our practice, ELISA is used for initial screening, with positive results confirmed by HEp-2 cell indirect immunofluorescence (IFA) upon request to determine ANA patterns. HEp-2 results are reported as titers based on serial dilutions.

A secondary outcome of this study was to explore the potential relationship between specific ANA characteristics—such as the titer and pattern—and all-cause mortality. ANA titers were assessed in a range from 1:40 to 1:640, with increments of twofold. The ANA patterns were analyzed according to the International Consensus on ANA patterns (ICAP) and included the speckled pattern (fine, AC-4 or coarse, AC-5), which is the standard in ANA serology testing at CHS, as well as centromere (AC-3), cytoplasmic (AC-15 to AC-28), dense fine speckled (DFS, AC-2), homogeneous (AC-1), mitochondrial (AC-21), mixed (more than one code), and other nuclear patterns (including AC-3 and AC-6 to AC-14).

Given the established association between ANA serology and systemic autoimmune diseases, an additional secondary outcome of this study was to investigate whether ANA positivity is significantly linked to increased mortality in patients diagnosed with various rheumatological conditions. These were identified using recorded diagnoses from the International Classification of Diseases, 9th Revision (ICD-9) [15], and included systemic lupus erythematosus (SLE, code 710.0), systemic sclerosis (diffuse type, code 710.1), Sjögren’s syndrome (code 710.2), rheumatoid arthritis (RA, code 714.0), and myositis (code 729.1).

In addition, we evaluated rheumatic diseases where ANA is not considered to play a pathogenic role—specifically, psoriatic arthritis (PsA, code 696.0), reactive arthritis (ReA, code 711.1), and ankylosing spondylitis (AS, code 720.0). All the rheumatological conditions analyzed in this study, regardless of whether ANA was involved as a major pathogenic factor, were categorized collectively as Rheumatologic-Immune Related Disorders (RIRD).

An asymptomatic ANA patient was defined as any patient with a positive ANA titer of at least 1:40 who did not have an additional diagnosis of any rheumatological disorders at the time of analysis (i.e., in 2023) included in our data.

The study also considered demographic parameters such as age (in years), gender (male or female), and ethnicity (classified as Arab or Jewish). Smoking status (current or past) was recorded. Metabolic comorbidities, including diabetes mellitus (both type 1 and type 2), hyperlipidemia, hypertension, and obesity, were documented, along with cancer-related comorbidities.

Statistical Analysis

Descriptive statistics were used to analyze the attributes of the study population. Each variable was presented using the most appropriate central and dispersion measures: nominal variables (e.g., patient sex, comorbidities) were presented as the number and percentage (%), while numerical (continuous) variables (e.g., patient age) were presented as the mean ± standard deviation (SD). The normality of continuous variables was assessed using histograms, Q-Q plots, the Shapiro–Wilk test, and the Kolmogorov–Smirnov test.

Initially, we conducted a univariate analysis to evaluate the clinical and sociodemographic characteristics of the study cohort, stratified by ANA status, and to assess their association with all-cause mortality among ANA-positive patients. For continuous variables, the Mann–Whitney test was used due to their non-normal distribution, and for nominal variables, the Chi-square test was employed. Next, we examined the association between ANA characteristics (e.g., ANA status, pattern, and titer) and all-cause mortality using Kaplan–Meier curves and the Log-rank test. Finally, we used multivariable Cox regression to assess the hazard ratio (HR) of patients’ ANA characteristics, adjusting for potential confounders. All analyses were conducted using SPSS Statistics V. 25 and R software version 4.2. A two-sided test significance level of 0.05 was maintained throughout the entire study.

Ethical Considerations

The study was approved by the Ethical Committee of Clalit Health Services (CHS) and was exempted from the requirement to obtain informed consent. The approval code is COM2-0212-17, and the approval date is 20 March 2022.

## 3. Results

All-cause mortality in ANA-positive vs. ANA-negative patients
▪Population

The study included 205,862 patients, evenly divided into two groups: 102,931/205,862 patients (50%) tested positive for ANA (study group), and 102,931/205,862 patients (50%) tested negative for ANA (control group). Table 1 outlines the baseline characteristics of each group. Patients with positive ANA serology were older than those with negative ANA serology, with a mean age of 53.2 years (SD ± 18.5) versus 47.4 years (SD ± 18.3) (*p* < 0.001). The prevalence of Jewish ethnicity was lower in the ANA-positive group compared to the ANA-negative group (84.1% vs. 88.0%, *p* < 0.001). Gender distribution (male/female) was similar between the two groups.

Regarding comorbidities, hypertension was more prevalent in the ANA-positive group compared to the ANA-negative group (40.0% vs. 38.7%, *p* < 0.001). Conversely, hyperlipidemia was less prevalent in the ANA-positive group compared to the ANA-negative group (58.3% vs. 59.3%, *p* < 0.001). The prevalence of autoimmune conditions (SLE, RA, diffuse systemic sclerosis, Sjögren’s, and myositis) was significantly higher in the ANA-positive group. As expected, the prevalence of spondyloarthropathies (ReA, PsA, AS) was lower in the ANA-negative group or statistically comparable between the two groups.

The overall prevalence of malignancy was significantly higher in the ANA-positive group compared to the ANA-negative group (17.4% vs. 16.7%, *p* < 0.001). However, the distribution of most malignancy subtypes was similar between the groups, with predominantly insignificant differences.


▪All-cause mortality


Figure 1 illustrates the progressively decreasing survival rates among ANA-positive patients compared to ANA-negative patients (*p* < 0.001). To identify specific predictors of all-cause mortality, a multivariate analysis was performed to evaluate potential associations between clinical and epidemiological factors and mortality risk (Appendix A). The hazard ratio (HR) for the association between positive ANA status and all-cause mortality was 6.18 (95% CI 6.0–6.3, *p* < 0.001). After adjusting for significant baseline variables, the adjusted HR (aHR) was 4.62 (95% CI 4.5–4.7, *p* < 0.001).

The presence of RIRDs was associated with increased all-cause mortality in ANA-positive patients, with a preliminary hazard ratio (HR) of 1.27 (95% CI, 1.22–1.32, *p* < 0.001). However, after adjusting for significant variables, there was a notable shift; the presence of RIRDs exhibited a relative protective effect, resulting in an adjusted hazard ratio (aHR) of 0.83 (95% CI, 0.79–0.86, *p* < 0.001). Other examined factors, such as age at ANA testing, Jewish ethnicity, hypertension, and malignancy, were significantly associated with all-cause mortality both before and after adjustment (*p* < 0.001).


▪ANA characteristics as mortality predictors


Table 2 illustrates the relationship between specific ANA characteristics and all-cause mortality.

Different ANA titers were found to correlate with all-cause mortality, with the strongest correlation observed at the lowest titer of 1:40, which had an aHR of 4.70 (95% CI 4.53–4.88, *p* < 0.001). Figure 2A visually represents the different survival curves for each ANA titer that was analyzed. Additionally, different ANA patterns were analyzed, and similar to the ANA titre analysis, all patterns were found to be significantly correlated with all-cause mortality. The highest associations were observed for the mitochondrial (AC-21), dense fine speckled (DFS) (AC-2), and cytoplasmic patterns (AC-15 to AC-28), with adjusted hazard ratios (aHRs) of 36.14 (95% CI 29.78–43.85, *p* < 0.001), 29.77 (95% CI 26.58–33.34, *p* < 0.001), and 20.97 (95% CI 18.77–23.43, *p* < 0.001), respectively. Figure 2B illustrates the survival curves for the different ANA patterns.

All-cause mortality in rheumatological-associated ANA vs. asymptomatic ANA patients

Given the established link between positive ANA serology and various rheumatological conditions, we evaluated the association of ANA positivity in specific subgroups of patients with confirmed rheumatological diagnoses, comparing them to those without these conditions. Table 3 outlines the baseline characteristics of the ANA-positive subgroup, while Figure 3 visually depicts the survival curve comparing patients with RIRD-associated ANA to those with asymptomatic ANA (aHR 0.9, 95% CI 0.86–0.95, *p* < 0.001).

In a sub-analysis of specific RIRDs, classic ANA-positive diseases, including SLE, systemic sclerosis (SS), and myositis, were statistically associated with increased mortality. SLE was found to be present in 769/14,083 patients (5.5%) who died before the end of study period, compared to 4045/88,848 patients (4.5%) who were alive (*p* < 0.001). SS was found in 370/14,083 patients (2.6%) who died before the end of study period, compared to 935/88,848 patients (1.1%) who were alive (*p* < 0.001). Myositis was found in 34/14,083 patients (0.2%) who died, compared to 88/88,848 patients (0.1%) who were alive (*p* < 0.001). In contrast, Sjögren’s syndrome and rheumatoid arthritis (RA) were not significantly correlated with mortality. RA was found in 585/14,083 patients (4.2%) who died, compared to 3409/88,848 patients (3.8%) who were alive (*p* = 0.07), and Sjogren’s was present in 273/14,083 patients (1.9%) who died, compared to 1742/88,848 patients (2%) who were alive (*p* = 0.86).

In this study, we also included rheumatological conditions driven by innate immunity or autoinflammatory processes, where ANA is not significant (e.g., PsA, ReA, AS). Most ANA-positive PsA patients had low titers of 1:40 (791/839, 94.2%), as did ANA-positive AS patients (107/132, 81.1%). PsA was negatively correlated with overall mortality, present in 802/88,848 patients (0.9%) who were alive by the end of the study, compared to 37/14,083 patients (0.3%) who died (*p* < 0.001). AS and ReA were not significantly correlated with mortality in ANA-positive patients.

## 4. Discussion

Antinuclear antibodies (ANAs) target nuclear proteins and are a key screening tool for diagnosing chronic inflammatory conditions, especially autoimmune diseases like SLE, Systemic sclerosis, and Sjogren’s syndrome. While ANA is sensitive for these diseases, it lacks specificity, as positive results also occur in healthy individuals and those with chronic infections, cancers, or medication-related effects [10,16].

To our knowledge, this is the largest cohort study to examine the relationship between ANA positivity and overall mortality. The main finding of our study is the association between positive ANA serology and all-cause mortality in a randomly tested large cohort of 205,862 patients, with an aHR of 6.18 when comparing patients with positive ANA to those with negative ANA.

At this point, it is important to assert that although an ANA titer of 1:80 is widely recognized as clinically significant, in this study, we included individuals with titers of 1:40 to maximize the cohort size and better analyze the association between ANA positivity and mortality in healthy individuals and those with autoimmune diseases.

The association between ANA and all-cause mortality in the general population has been explored in the past, but the evidence remains limited. A study using data from the National Health and Nutrition Examination Survey (NHANES) from 1999 to 2004, which included 3357 adults with ANA measurements and mortality data through 2011, found that ANA was not strongly associated with all-cause mortality (HR of 1.13, *p* = 1.60) [4]. However, in subgroups with a history of cancer, positive ANA was associated with elevated all-cause mortality, particularly in men and older participants (≥75 years) [4]. In a study by Hurme et al., ANA positivity did not affect survival rates or inflammation levels in nonagenarians, suggesting that ANA positivity alone, without an accompanying autoimmune condition, does not significantly impact mortality [17]. Conversely, in a retrospective study by Selmi et al., which evaluated data from a random cohort of 2822 patients in Italy over a 15-year period, there was initially a reduced unadjusted survival in subjects with a positive ANA serology. However, this significance was lost after adjusting for age and sex [1]. Our study, which used a comparable follow-up period in a randomly assigned patient cohort, identified a clear and significant correlation between ANA positivity and mortality. This finding may be attributed to the larger cohort size in our study.

In the specific context of malignancy in patients with positive ANA, some studies suggest that ANA positivity may be linked to an increased risk of cancer and higher mortality, leading to a poorer prognosis. However, the evidence remains mixed. For example, a 15-year study found no significant association between ANA levels and elevated cancer risk or mortality [1]. Conversely, another study reported a significant correlation between the presence of ANA and survival rates, though no connection was found between ANA positivity and tumor stage, histological type, or the intensity and pattern of ANA staining [13]. In our study, since malignancy was statistically significantly more prevalent in the ANA-positive group compared to the ANA-negative group, we attempted to adjust for the impact of malignancy. However, the correlation between ANA positivity and mortality persisted, with an adjusted hazard ratio (aHR) of 4.62. This finding suggests that the association between ANA and overall mortality is independent of any coexisting malignancy.

Aiming to identify specific predictors of all-cause mortality, we performed a multivariate regression analysis, adjusting for significant baseline variables. This adjustment resulted in a shift in the HR for rheumatological conditions, transforming them from a risk factor to a protective factor after accounting for ANA status and several other parameters. Given the prevalence of ANA positivity in both rheumatological conditions and the healthy population, we conducted a sub-analysis to assess the impact of rheumatological conditions on mortality in ANA-positive patients.

Over a 14-year follow-up period, we found that patients with rheumatological-associated ANA had a higher survival rate compared to those with asymptomatic ANA. Previous studies, particularly those focused on SLE, suggest that patients with known chronic inflammatory condition have more rigorous monitoring, regular check-ups, and supplementary examinations [18,19]. This heightened vigilance can facilitate the early detection of other potential causes of mortality. For instance, Yazdany et al. found that frequent physician visits and generalist involvement improved preventive care in SLE including cancer screening and immunizations, highlighting the importance of regular monitoring and comprehensive management [19]. Additionally, a study by Castro et al. demonstrated that SLE patients had higher screening rates for several metabolic and non-metabolic conditions, including hypertension, diabetes, and osteoporosis, compared to non-SLE counterparts [18]. The authors concluded that SLE patients are more likely to receive certain preventive services, potentially leading to the earlier detection and management of comorbid conditions [18]. Furthermore, a study by Bruera et al. found that while cervical cancer screening rates were higher among women with SLE compared to the general population, a significant portion of SLE patients still did not undergo screening [20]. The study concluded that although SLE patients are more likely to receive certain preventive services, there remains a need for improvement in providing comprehensive care [20].

Our study found lower survival rates in asymptomatic ANA patients compared to those with rheumatological-associated ANA. We hypothesize that one potential explanation is the less frequent monitoring of asymptomatic patients, which may delay the diagnosis of non-inflammatory but life-threatening conditions, such as malignancies or severe cardiovascular diseases. However, another possible explanation is that some patients may have atypical presentations of autoimmune or autoinflammatory conditions that go unrecognized, leading to underdiagnosis or delayed diagnosis. The observation that the survival curves in Figure 3 begin to diverge at the five-year mark supports the first hypothesis, suggesting that mortality may be more closely linked to additional comorbidities rather than the ANA-associated condition itself. Searching the relevant literature, various studies show that delays in diagnosing chronic inflammatory conditions, like RA-ILD [21] and SLE [22], increase mortality and disease-related damage [21,22].

Our findings raise concerns about potential underdiagnosis, which may contribute to increased morbidity and mortality rates in asymptomatic ANA-positive patients. While these results suggest the need for tailored follow-up protocols for this population, we acknowledge that, in a broader context, the cost-effectiveness of such approaches may be limited.

In addition, our sub-analysis revealed that different ANA patterns vary with the increased mortality risk. Among the ANA patterns, the mitochondrial pattern (AC-21) exhibited the highest HR: (36.14, 95% CI: 29.78–43.85, *p* < 0.001), followed by the DFS pattern (AC-2) (HR: 29.77, 95% CI: 26.58–33.34, *p* < 0.001), and the cytoplasmic pattern (AC-15 to AC-28) (HR: 20.97, 95% CI: 18.77–23.43, *p* < 0.001). These patterns, which indicate specific types of autoantibodies, may reflect more severe systemic involvement or pathogenic processes, thus contributing to higher mortality. For instance, the mitochondrial pattern is mainly associated with primary biliary cirrhosis and the presence of anti-mitochondrial antibodies [23], with patients at a higher risk of malignancies, particularly hepatocarcinoma, but also lymphomas and other gastrointestinal cancers, which may explain the elevated mortality rate in these patients [24]. Surprisingly, the DFS (AC-2) pattern had the second-highest mortality rate. Although DFS (AC-2) is less commonly linked with traditional autoimmune diseases [25] and can often be found in healthy individuals [26], its strong association with increased mortality in our study may be explained by the role of its antigen (DFS70/LEDGFp75) as a stress protein, which potentially indicates cellular stress and inflammation and could contribute to a higher mortality risk [27]. This pattern has also been observed in non-autoimmune conditions, including malignancies, and is the most common among breast and prostate cancer patients with positive ANA.

An analysis of specific ANA titers revealed that while all titer levels were independently associated with increased mortality risk, higher titers did not always correspond to the highest hazard ratios. For example, a titer of 1:160 had a higher hazard ratio (HR: 4.17, 95% CI: 3.94–4.41, *p* < 0.001) compared to 1:320 (HR: 3.72, 95% CI: 3.19–4.33, *p* < 0.001) and 1:640 (HR: 3.64, 95% CI: 2.93–4.52, *p* < 0.001). This suggests that while higher titers reflect significant autoimmune activity, lower titers may be linked to higher mortality, possibly due to underdiagnosis, undertreatment, or associations with tissue damage, inflammation, or malignancies. This raises the possibility that lower titers may not receive the same level of clinical attention, leading to missed opportunities for early intervention. It is important to assert that a key limitation of this observation is that ANA titers are typically measured at a single time point, leaving the peak titer unknown, especially if titers fluctuate over time. Nonetheless, these findings underscore the importance of comprehensive ANA characterization, highlighting that both the titer and pattern of ANA may play a role in mortality risk assessment.

This study has several limitations. In our opinion, the most significant limitation is the lack of data on cause-specific mortality, such as cardiovascular causes and liver disease. However, as this is the first study to address this question based on large cohort analysis, we think that it still offers valuable insights. Other limitations of this study include reliance on registry-based diagnoses and the inability to establish causal relationships. We also lacked detailed information on therapies, such as anti-TNF biologics, hydralazine, and treatments for autoimmune conditions like thyroiditis, which may be linked to ANA positivity. Nonetheless, the CHS registry’s diagnostic accuracy is well-regarded, with systematic validation and cross-verification across multiple data sources.

## 5. Conclusions

This is the first large-scale analysis of mortality in ANA-positive versus ANA-negative patients. ANA positivity, especially in asymptomatic individuals, is linked to increased mortality, even after adjusting for age, highlighting the need for improved screening. Further research is needed to confirm these findings.

## Figures and Tables

**Figure 1 medicina-61-00060-f001:**
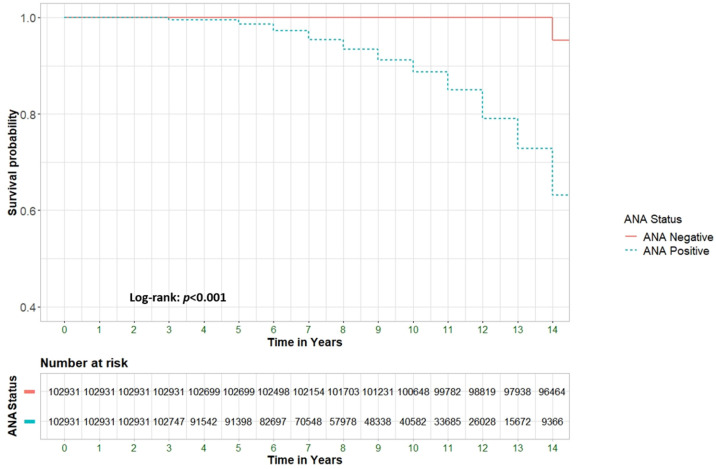
Survival analysis of patients with positive ANA compared to patients with negative ANA.

**Figure 2 medicina-61-00060-f002:**
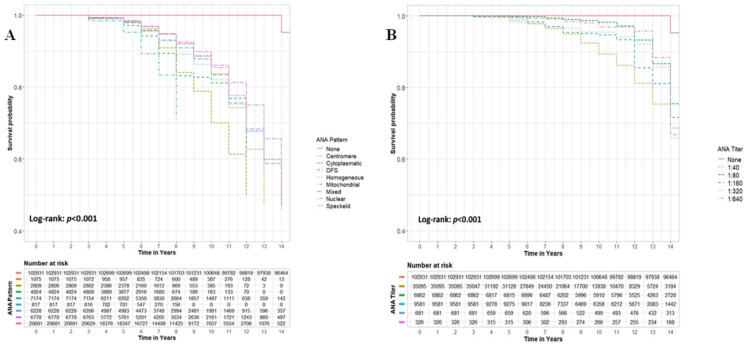
Survival rates among different ANA patterns (**A**) and different ANA titers (**B**).

**Figure 3 medicina-61-00060-f003:**
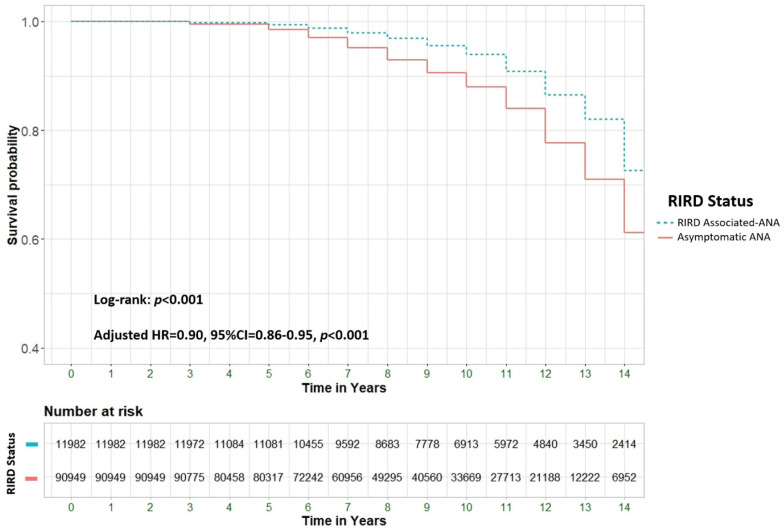
Survival rates among patients with positive RIRD (Rheumatological immune-related disorder) status compared to asymptomatic patients. HR = Hazard Ratio, CI = Confidence Interval. Cox Regression adjusted to age, ethnicity, sex, and comorbidities.

**Table 1 medicina-61-00060-t001:** Clinical and sociodemographic characteristics of the Study Cohort stratified by ANA Status.

Variable		All Patients	ANA Positive	ANA Negative	*p*-Value
		(N = 205,862)	(N = 102,931)	(N = 102,931)	
Patient Sex	Male	54,406 (26.4)	27,203 (26.4)	27,203 (26.4)	1.00
	Female	151,456 (73.6)	75,728 (73.6)	75,728 (73.6)	
Age at ANA test, years		50.3 ± 18.6	53.2 ± 18.5	47.4 ± 18.3	**<0.001**
Ethnicity	Jewish	176,850 (86.0)	86,448 (84.1)	90,402 (88.0)	**<0.001**
	Arab	28,701 (14.0)	16,334 (15.9)	12,367 (12.0)	
Smoking		72,363 (35.2)	36,305 (35.3)	36,058 (35.0)	0.254
Comorbidities	Diabetes Mellitus	49,689 (24.1)	24,858 (24.2)	24,831 (24.1)	0.889
	Hyperlipidemia	121,077 (58.8)	60,028 (58.3)	61,049 (59.3)	**<0.001**
	Hypertension	80,934 (39.3)	41,138 (40.0)	39,796 (38.7)	**<0.001**
	Obesity	66,757 (32.4)	33,540 (32.6)	33,217 (32.3)	0.128
Rheumatological immune-related disorders (RIRD)	Any	16,429 (8.0)	11,982 (11.6)	4447 (4.3)	**<0.001**
	SLE	5063 (2.5)	4445 (4.3)	618 (0.6)	**<0.001**
	Rheumatoid arthritis	6193 (3.0)	3994 (3.9)	2199 (2.1)	**<0.001**
	Systemic Sclerosis (diffused)	1191 (0.6)	1113 (1.1)	78 (0.1)	**<0.001**
	Sjogren	2500 (1.2)	2015 (2.0)	485 (0.5)	**<0.001**
	Myositis	144 (0.1)	122 (0.1)	22 (0.0)	**<0.001**
	Reactive arthritis	22 (0.0)	6 (0.0)	16 (0.0)	**0.033**
	Psoriatic arthritis	1888 (0.9)	839 (0.4)	1049 (0.9)	**<0.001**
	Ankylosing Spondylitis	919 (0.4)	132 (0.1)	787 (0.9)	**<0.001**
Malignancy	Any	35,030 (17.0)	17,864 (17.4)	17,166 (16.7)	**<0.001**
	Solid	29,613 (14.4)	14,995 (14.6)	14,618 (14.2)	**0.018**
	Hematologic	5417 (2.6)	2869 (2.8)	2548 (2.5)	**<0.001**
	Breast	7360 (3.6)	3602 (3.5)	3758 (3.7)	0.064
	CRC	3468 (1.7)	1768 (1.7)	1700 (1.7)	0.244
	Prostate	1995 (1.0)	967 (0.9)	1028 (1.0)	0.170
	Lung	1863 (0.9)	991 (1.0)	872 (0.8)	**0.006**
	Bladder	1515 (0.7)	781 (0.8)	734 (0.7)	0.226
	Ovary	611 (0.3)	329 (0.3)	282 (0.3)	0.057
	Uterus	1063 (0.5)	509 (0.5)	554 (0.5)	0.166
	Pancreas	651 (0.3)	313 (0.3)	338 (0.3)	0.326
	CNS	459 (0.2)	216 (0.2)	243 (0.2)	0.207
	Stomach	596 (0.3)	306 (0.3)	290 (0.3)	0.512
	Melanoma	1853 (0.9)	904 (0.9)	949 (0.9)	0.294
	HL	546 (0.3)	310 (0.3)	236 (0.2)	**0.002**
	NHL	2270 (1.1)	1251 (1.2)	1019 (1.0)	**<0.001**
	AL	462 (0.2)	228 (0.2)	234 (0.2)	0.780
	CL	556 (0.3)	286 (0.3)	270 (0.3)	0.497
	Kidney	908 (0.4)	457 (0.4)	451 (0.4)	0.842
	Larynx	336 (0.2)	174 (0.2)	162 (0.2)	0.512
	Cervix	1112 (0.5)	590 (0.6)	522 (0.5)	**0.041**
	Pharynx	424 (0.2)	232 (0.2)	192 (0.2)	0.052
	Esophagus	70 (0.0)	34 (0.0)	36 (0.0)	0.811
	Liver / Bile	512 (0.2)	281 (0.3)	231 (0.2)	**0.027**
	Thyroid	1620 (0.8)	820 (0.8)	800 (0.8)	0.618
	Bone	110 (0.1)	59 (0.1)	51 (0.0)	0.445
	Sarcoma	490 (0.2)	249 (0.2)	241 (0.2)	0.717
	Genital	264 (0.1)	143 (0.1)	121 (0.1)	0.175
	MM	706 (0.3)	348 (0.3)	358 (0.3)	0.706
	PV	479 (0.2)	231 (0.2)	248 (0.2)	0.437
	MDS	287 (0.1)	154 (0.1)	133 (0.1)	0.215
	MPS	111 (0.1)	61 (0.1)	50 (0.0)	0.296
	Other Site	1071 (0.5)	536 (0.5)	535 (0.5)	0.976
	Unknown Site	1262 (0.6)	734 (0.7)	528 (0.5)	**<0.001**

**Table 2 medicina-61-00060-t002:** Association Between ANA Characteristics and All-Cause Mortality.

Variable		Adjusted HR ^b^	95% CI	Pv
**Positive ANA Antibody**		4.79	4.66–4.92	**<0.001**
ANA Pattern	No ANA Antibody	Reference		
	Centromere (AC3)	17.54	15.00–20.51	**<0.001**
	Cytoplasmatic (AC-15 to AC-23)	20.97	18.77–23.43	**<0.001**
	DFS (AC-2)	29.77	26.58–33.34	**<0.001**
	Homogenous (AC-1)	12.95	12.03–13.94	**<0.001**
	Mitochondrial (AC-21)	36.14	29.78–43.85	**<0.001**
	Mixed	9.71	9.08–10.37	**<0.001**
	Nuclear (AC-3, AC-6 to AC-14).	8.06	7.54–8.61	**<0.001**
	Speckled (AC-4, AC-5)	12.54	11.95–13.15	**<0.001**
ANA Titer	No ANA Antibody			
	1:40	4.7	4.53–4.88	**<0.001**
	0.097222	2.54	2.40–2.69	**<0.001**
	0.152778	4.17	3.94–4.41	**<0.001**
	0.263889	3.72	3.19–4.33	**<0.001**
	0.486111	3.64	2.93–4.52	**<0.001**

Note: Boldface type indicates *p* < 0.05. HR = Hazard Ratio; CI = Confidence Interval; Pv = *p*-value. ^b^ Multivariable Cox Regression, adjusted to patient age, sex, ethnicity, smoking history, autoimmune disease, diabetes mellites, hyperlipidemia, hypertension, obesity, and malignancy.

**Table 3 medicina-61-00060-t003:** Clinical and sociodemographic characteristics Associated with All-Cause Mortality Among ANA-Positive Patients.

Variable		All Patients	Died	Alive	*p*-Value
		(N = 102,931)	(N = 14,083)	(N = 88,848)	
Patient Sex	Male	27,203 (26.4)	5514 (39.2)	21,689 (24.4)	**<0.001**
	Female	75,728 (73.6)	8569 (60.8)	67,159 (75.6)	
Age at ANA Test, years		53.2 ± 18.5	72.4 ± 13.0	50.1 ± 17.3	**<0.001**
Ethnicity	Jewish	86,448 (84.1)	12,609 (89.6)	73,839 (83.2)	**<0.001**
	Arab	16,334 (15.9)	1466 (10.4)	14,868 (16.8)	
Smoking		36,305 (35.6)	5920 (42.0)	30,385 (34.2)	**<0.001**
Comorbidities	Diabetes Mellitus	24,858 (24.2)	6450 (45.8)	18,408 (20.7)	**<0.001**
	Hyperlipidemia	60,028 (58.3)	11,490 (81.6)	48,538 (54.6)	**<0.001**
	Hypertension	41,138 (40.0)	11,167 (79.3)	29,971 (33.7)	**<0.001**
	Obesity	33,540 (32.6)	5458 (38.8)	28,082 (31.6)	**<0.001**
Rheumatological immune-related disorders (RIRD).	Any	11,982 (11.6)	1684 (12.0)	10,298 (11.6)	0.207
	SLE	4814 (4.6)	769 (5.5)	4045 (4.5)	**<0.001**
	Psoriatic arthritis	839 (0.8)	37 (0.3)	802 (0.9)	**<0.001**
	Systemic sclerosis (diffuse)	1305 (1.3)	370 (2.6)	935 (1.1)	**<0.001**
	Sjogren	2015 (2.0)	273 (1.9)	1742 (2.0)	0.860
	Myositis	122 (0.1)	34 (0.2)	88 (0.1)	**<0.001**
	Reactive arthritis	6 (0.0)	1 (0.0)	5 (0.0)	0.832
	Rheumatoid arthritis	3994 (3.9)	585 (4.2)	3409 (3.8)	0.070
	Ankylosing Spondylitis	132 (0.1)	54 (0.1)	115 (0.1)	**0.424**
Malignancy	Any	17,864 (17.4)	6630 (47.1)	11,234 (12.6)	**<0.001**

## Data Availability

The datasets generated and/or analyzed during the current research are available from the corresponding author on reasonable request.

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
