# Peer review of "Mortality in Antinuclear Antibody-Positive Patients with and Without Rheumatologic Immune-Related Disorders: A Large-Scale Population-Based Study"

_medicina, 2025, doi:10.3390/medicina61010060_

Round 1
Reviewer 1 Report
Comments and Suggestions for Authors
The article is interesting, with a population-based database with a very large sample and long follow-up period. It is well written and structured. I suggest some modifications that I believe would improve the understanding of the study:
1.Abstract/Results. Better to show HRs from cox regression than data from table 1 between groups.
2. Methods. Anti-mitochondrial is looked at but no detail on primary biliary cirrhosis data between groups. Data may not be available but it should be added if possible.
3. Methods/outcomes/fifth paragraph: correct typo at the end of the sentence.
4. Results/1st and 2nd paragraphs: always compare the 2 groups in the same order, otherwise the reading becomes a bit chaotic. For example ANA + vs. ANA -. I think it is more visual to show it as (.....% vs. ....%, p....).
5. Table 1. Show autoimmune diseases in the same order as in the text. I miss comorbidities that are important for the study.
6. Table 2/discussion. The ANA titers do not parallel the increase in HR as you say. It would certainly have been more visual. One possible explanation is that the ANA determination is at a certain time in the patient's life but that does not mean that the titers will increase or decrease in the future. The peak titer of each patient is actually unknown. To be added for discussion if appropriate.
7. Discussion/1st paragraph. I would name here the same autoimmune diseases that are discussed throughout the text. If you wish to introduce others such as MCTD you will have to define the acronyms.
8. Discussion. You stated that ANA + asymptomatic have more mortality than those with rheumatologic disease and it is suggested that this is due to less follow-up. Perhaps Figure 3 in which it is observed that the curves separate at 5 years indicates that in reality this mortality is not so much related to the disease associated with ANA (perhaps it would have had an earlier mortality) but to the follow-up as you suggest. On the other hand, do the authors suggest the creation of specific follow-up for these patients? It exists in some hospitals, although its cost-effectiveness may not be high. I would make some comments in this direction and suggest how the follow-up should be (periodicity, clinical vs. analytical...).
9. Table 2/supplemental: I don't quite understand why there are 2 regressions with somewhat different variables included. Unify in only 1 regression that would be the definitive table 2 and eliminate the supplementary one. I think it is important to introduce the results of all the variables and not only comment at the bottom of the table the variables introduced in the model. Making a population mortality study without including heart disease (1st cause of mortality in many countries), dementia, renal failure or liver disease is an important limitation of the results. It is possible that these data are recorded in this database; I would recommend their inclusion. In my opinion this is the biggest weakness of this study.
10. In results I would make different sections for each study because otherwise it is confusing.
a. ANA + vs ANA -: study population/risk for all-cause mortality/table 1-2/Figure 1-2
b. ANA+ with rheumathological disease vs ANA + assymptomatic: study population/risk for all-cause mortality/table 3/figure 3
Author Response
Answers to reviewers’ comment - report 1
- Reviewer: Abstract/Results. Better to show HRs from cox regression than data from table 1 between groups.
- Answer: Thank you for the comment. The current abstract results section includes HRs from cox regression.
- Reviewer: Anti-mitochondrial is looked at but no detail on primary biliary cirrhosis data between groups. Data may not be available but it should be added if possible.
- Answer: Thank you for this importatn point. unfortunarely, we did not have data about primary biliary cirhhosis between the group.
- Reviewer: Methods/outcomes/fifth paragraph: correct typo at the end of the sentence.
- Answer: Thank you for this comment. Corrected in the manuscript.
- Reviewer: Results/1st and 2nd paragraphs: always compare the 2 groups in the same order, otherwise the reading becomes a bit chaotic. For example ANA + vs. ANA -. I think it is more visual to show it as (.....% vs. ....%, p....).
- Answer: Thank you for this comment. Corrected in the manuscript.
- Reviewer: Table 1. Show autoimmune diseases in the same order as in the text. I miss comorbidities that are important for the study.
- Answer: Thank you for highlighting this important point. The paragraph has been revised in the text to align with the order presented in the table. Regarding other autoimmune comorbidities, unfortunately, the diseases listed in the table are the only ones available in our original large database.
- Reviewer: Table 2/discussion. The ANA titers do not parallel the increase in HR as you say. It would certainly have been more visual. One possible explanation is that the ANA determination is at a certain time in the patient's life but that does not mean that the titers will increase or decrease in the future. The peak titer of each patient is actually unknown. To be added for discussion if appropriate.
- Answer: Thank you for highlighting this important point. The paragraph has been revised in the discussion.
- Reviewer: Discussion/1st paragraph. I would name here the same autoimmune diseases that are discussed throughout the text. If you wish to introduce others such as MCTD you will have to define the acronyms.
- Answer: Thank you for this comment. The text was changed.
- Reviewer: You stated that ANA + asymptomatic have more mortality than those with rheumatologic disease and it is suggested that this is due to less follow-up. Perhaps Figure 3 in which it is observed that the curves separate at 5 years indicates that in reality this mortality is not so much related to the disease associated with ANA (perhaps it would have had an earlier mortality) but to the follow-up as you suggest. On the other hand, do the authors suggest the creation of specific follow-up for these patients? It exists in some hospitals, although its cost-effectiveness may not be high. I would make some comments in this direction and suggest how the follow-up should be (periodicity, clinical vs. analytical...).
- Answer: Thank you for this interesting point. We added relevant comments in the discussion section.
- Reviewer: Table 2/supplemental: I don't quite understand why there are 2 regressions with somewhat different variables included. Unify in only 1 regression that would be the definitive table 2 and eliminate the supplementary one. I think it is important to introduce the results of all the variables and not only comment at the bottom of the table the variables introduced in the model. Making a population mortality study without including heart disease (1st cause of mortality in many countries), dementia, renal failure or liver disease is an important limitation of the results. It is possible that these data are recorded in this database; I would recommend their inclusion. In my opinion this is the biggest weakness of this study.
- Answer: Thank you for this comment.
- First, Supplementary Table 1 presents the multivariate analysis aimed at identifying specific predictors of all-cause mortality in ANA-positive patients after adjustment. This adjustment led to a shift in the hazard ratios (HRs) for rheumatological conditions, transforming them from risk factors to protective factors after accounting for ANA status and other parameters. We found this shift particularly interesting, prompting us to hypothesize and explore further the specific role of rheumatological immune-related diseases (RIRD) in the ANA-positive patient population.
- Second, we acknowledge this significant limitation of the study. Unfortunately, data on cause-specific mortality, particularly for cardiovascular causes, dementia, and others, were not available. However, given that this is the first study to address this question with such a large cohort, we believe it still holds considerable strengths. As the above comment is more than relevant, we updated the limitations section.
- Answer: Thank you for this comment.
- Reviewer: In results I would make different sections for each study because otherwise it is confusing.
- ANA + vs ANA -: study population/risk for all-cause mortality/table 1-2/Figure 1-2
- ANA+ with rheumathological disease vs ANA + assymptomatic: study population/risk for all-cause mortality/table 3/figure 3
- Answer: Thank you for this comment. We changed the relevant sections.

Reviewer 2 Report
Comments and Suggestions for Authors
- The registration number and approval from the ethics committee must be included.
- Previous reports have described patients with infectious diseases, such as COVID-19, and complications like Long-COVID testing positive for ANA. This aspect should be included and discussed, especially considering that the study period includes the time with the highest mortality by SARS-CoV-2.
- It is necessary to justify why patients with ANA titers greater than 1:40 were considered positive, given that clinical relevance, particularly in rheumatic diseases, is well-documented for titers starting at 1:80.
- Information regarding the methodology used should be provided. Is necessary to clarify if the patient's samples were evaluated by a quantitative or qualitative ELISA, and detail the technique used to determine the ANA pattern, was using indirect immunofluorescence, ELISA, or both.
- It is not clear why ANA titers were established based on dilutions and not the quantitative value. This could be clarified better by defining the methodology used.
- The authors mention that the retrospective study involves the years 2000–2022; however, the starting year of record collection is not specified. This information is crucial, as the study aims to analyze mortality, and it is necessary to establish the timeframe during which these individuals passed away. Figure 1 suggests approximately 14 years, but this should be explicitly stated in the text, preferably in the methodology section.
- ANA patterns should be categorized and defined according to ICAP criteria.
- Table 3 should include data distinguishing cancer patients who survived from those who died.
Author Response
- Reviewer: The registration number and approval from the ethics committee must be included.
- Answer: Thank you for this comment. Approval number and date were added to the manuscript under methods section.
- Reviewer: Previous reports have described patients with infectious diseases, such as COVID-19, and complications like Long-COVID testing positive for ANA. This aspect should be included and discussed, especially considering that the study period includes the time with the highest mortality by SARS-CoV-2.
- Answer: Thank you for raising this important point. This study includes a large cohort of over 200,000 patients. However, unforunately, we did not have available and concise data regarding COVID-19 comorbidities, COVID-19-associated hospitalizations, or complications such as long COVID.
- Reviewer: It is necessary to justify why patients with ANA titers greater than 1:40 were considered positive, given that clinical relevance, particularly in rheumatic diseases, is well-documented for titers starting at 1:80.
- Answer: Thank you for this comment. We agree that an ANA titer of 1:80 is of higher clinical sign However, for this study, our objective was to investigate the association of ANA positivity with mortality and to ensure a sufficiently large cohort. By including ANA-positive individuals starting at a titer of 1:40, we were able to maximize the number of participants and compare outcomes between the healthy population with low ANA positivity (1:40) and patients with autoimmune diseases. This approach allowed us to detect higher mortality rates in the ANA-positive healthy population, reinforcing the relevance of our decision to include the lower titer threshold. We have included this explanation in the revised manuscript, where now we state on page 16 :"Although an ANA titer of 1:80 is widely recognized as clinically significant, we opted to include individuals with ANA titers starting at 1:40 to maximize the cohort size and enable a more comprehensive analysis, particularly to explore the association of ANA positivity with mortality in both healthy individuals and patients with autoimmune diseases."
- Reviewer: Information regarding the methodology used should be provided. Is necessary to clarify if the patient's samples were evaluated by a quantitative or qualitative ELISA, and detail the technique used to determine the ANA pattern, was using indirect immunofluorescence, ELISA, or both.
- Answer: In our study, ANA testing was conducted exclusively using HEp-2 cell IFA for pattern identification and titer determination. ELISA is used only as a screening tool in our practice, and positive ELISA results are followed by HEp-2 testing when requested. This approach ensures consistency in pattern identification and reporting. We have updated the manuscript to clarify this methodology and have now added the following: “In our practice, ELISA is used for initial screening, and positive ELISA results are followed by HEp-2 cell indirect immunofluorescence (IFA) upon request to determine the ANA pattern. Results from HEp-2 testing are reported as titers based on serial dilutions.”
- Reviewer: It is not clear why ANA titers were established based on dilutions and not the quantitative value. This could be clarified better by defining the methodology used.
- Answer: In our practice, ELISA is used only for initial screening, and if negative, the test is considered negative. If the ELISA result is positive, HEp-2 testing is performed upon request to identify the ANA pattern. ANA performed by HEp-2 cells immunoflorescence in our practice are reported as titers, reflecting serial dilutions. Quantitative values are not provided in this context. This methodology has been clarified in the revised manuscript. “In our practice, ELISA is used for initial screening, and positive ELISA results are followed by HEp-2 cell indirect immunofluorescence (IFA) upon request to determine the ANA pattern. Results from HEp-2 testing are reported as titers based on serial dilutions.”
- Reviewer: The authors mention that the retrospective study involves the years 2000–2022; however, the starting year of record collection is not specified. This information is crucial, as the study aims to analyze mortality, and it is necessary to establish the timeframe during which these individuals passed away. Figure 1 suggests approximately 14 years, but this should be explicitly stated in the text, preferably in the methodology section.
- Answer: Thank you for raising this point. We have added a note under the methodology section.
- Reviewer: ANA patterns should be categorized and defined according to ICAP criteria.
- Answer: Thank you for pointing this out. We have revised the manuscript to categorize and define the ANA patterns according to the ICAP (International Consensus on ANA Patterns) criteria.
- Reviewer: Table 3 should include data distinguishing cancer patients who survived from those who died.
- Answer: Thank you for your comment. The last row of Table 3 provides the total number of patients with malignancy (17,864 patients), of whom 12.6% were alive, and 47.1% had died. These figures represent all-cause mortality. however, unfortunately, we lacked high-quality data on specific cancer-related deaths. The subsequent multivariate analysis, which evaluated mortality in RIRD-associated ANA patients compared to asymptomatic ANA patients, was adjusted for various parameters, including comorbidities and malignancies.

Round 2
Reviewer 1 Report
Comments and Suggestions for Authors
Thank you for the effort in correcting the manuscript. It would remain to correct only one typo in results/ANA + vs -/1st paragraph: the age of the groups is incorrect (there has been a confusion of columns).
Author Response
Comment: It would remain to correct only one typo in results/ANA + vs -/1st paragraph: the age of the groups is incorrect (there has been a confusion of columns).
Answer: Thank you for this comment. It was corrected it the updated manuscript.